# Multiscale Two-Stream Fusion Network for Benggang Classification in Multi-Source Images

**DOI:** 10.3390/s25092924

**Published:** 2025-05-06

**Authors:** Xuli Rao, Chen Feng, Jinshi Lin, Zhide Chen, Xiang Ji, Yanhe Huang, Renguang Chen

**Affiliations:** 1Jinshan Soil and Water Conservation Research Center, Fujian Agriculture and Forestry University, Fuzhou 350002, Chinalinjs18@163.com (J.L.); jixiangss@126.com (X.J.); 2School of Information Engineering, Fuzhou Polytechnic, Fuzhou 350108, China; fc@fvti.edu.cn; 3College of Computer and Cyber Security, Fujian Normal University, Fuzhou 350117, China; zhidechen@fjnu.edu.cn (Z.C.); qsx20221346@student.fjnu.edu.cn (R.C.)

**Keywords:** Benggang classification, multiscale features, two-stream fusion network, multi-source image fusion, attention mechanism

## Abstract

Benggangs, a type of soil erosion widely distributed in the hilly and mountainous regions of South China, pose significant challenges to land management and ecological conservation. Accurate identification and assessment of their location and scale are essential for effective Benggang control. With advancements in technology, deep learning has emerged as a critical tool for Benggang classification. However, selecting suitable feature extraction and fusion methods for multi-source image data remains a significant challenge. This study proposes a Benggang classification method based on multiscale features and a two-stream fusion network (MS-TSFN). Key features of targeted Benggang areas, such as slope, aspect, curvature, hill shade, and edge, were extracted from Digital Orthophotography Map (DOM) and Digital Surface Model (DSM) data collected by drones. The two-stream fusion network, with ResNeSt as the backbone, extracted multiscale features from multi-source images and an attention-based feature fusion block was developed to explore complementary associations among features and achieve deep fusion of information across data types. A decision fusion block was employed for global prediction to classify areas as Benggang or non-Benggang. Experimental comparisons of different data inputs and network models revealed that the proposed method outperformed current state-of-the-art approaches in extracting spatial features and textures of Benggangs. The best results were obtained using a combination of DOM data, Canny edge detection, and DSM features in multi-source images. Specifically, the proposed model achieved an accuracy of 92.76%, a precision of 85.00%, a recall of 77.27%, and an F1-score of 0.8059, demonstrating its adaptability and high identification accuracy under complex terrain conditions.

## 1. Introduction

A Benggang is a unique geomorphic erosion widely distributed in the red soil region of South China, primarily in the hilly and mountainous areas covered by weathered crusts, such as those in Guangdong, Fujian, Jiangxi, and other provinces south of the Yangtze River [1]. The formation of Benggangs is the result of the combined effects of gravity and hydrodynamics, which cause the collapse, transport, and deposition of slope soils or weathered crusts. A Benggang typically appears as a chair-shaped landform and generally consists of five main components [2]: catchment area, collapse slope, debris accumulation, gully, and alluvial fan. Figure 1 shows the core erosion unit of a Benggang—the collapsing wall and colluvial deposit area. This region is characterized by significant geomorphic fragmentation: the collapsing walls are steep and bare, the colluvial deposits are disordered and mixed, and the original vegetation system has severely degraded.

In addition to reducing land use efficiency and deteriorating the local ecological environment, Benggangs exacerbate natural disasters such as landslides and mudslides, posing significant threats to national security, food security, ecological stability, and public safety in the low hilly and mountainous areas throughout the red soil region in South China. Furthermore, they severely constrain the sustainable development of both local ecosystems and socio-economic systems [3,4]. Therefore, the identification and prevention of Benggangs is crucial to soil and water conservation and ecological management.

Scholars worldwide have conducted a series of studies on Benggang erosion using information technology. Duan and Lin conducted an in-depth analysis of factors influencing Benggang erosion through both overall surveys and typical surveys [5,6]. Liao et al. used GIS technology to locate Benggangs, determine boundaries, and analyze the relationship between geology and Benggang erosion and its spatial distribution [7]. In earlier studies, researchers manually analyzed remote-sensing images, identified suspected Benggang areas, and conducted field investigations in these areas. While methods allowed for precise identification of Benggangs in small areas, they were not capable of rapid detection and localization of Benggangs across large areas due to their reliance on field surveys and manual analyses. With the widespread adoption of automated drone measurement technologies, many researchers have built 3D models using drone aerial imagery data combined with GIS spatial analysis to monitor and survey Benggangs [8,9,10]. The results reveal that drone technology, with its shorter cycles and higher precision, offers significant advantages over traditional airborne remote sensing for data collection in small areas, demonstrating substantial application potential and practical value for Benggang monitoring in localized regions.

In recent years, breakthroughs in deep learning for computer vision have introduced a wide range of novel methods and tools for the interpretation of remote-sensing images. Achieving precise Benggang identification using modern remote sensing technology and deep learning algorithms has become an important research direction [11]. This not only improves the accuracy of Benggang monitoring but also significantly enhances identification speed compared to traditional manual identification methods. For instance, Shen et al. [12] generated a hybrid representation of Digital Orthophotography Map (DOM) and Digital Surface Model (DSM) features for Benggang areas with the Bag of Visual Topographical Words. They created high-level semantic representations through Latent Dirichlet Allocation (LDA) and finally used Support Vector Machine (SVM) for Benggang classification based on fused features. Hu et al. proposed a new method to detect Benggang areas from remote-sensing images with the Segment Anything Model (SAM) and Object-Based Classification (OBC) [13]. Shen et al. effectively improved the detection accuracy of Benggang areas by deeply fusing DOM and DSM data with a two-stream convolutional neural network model [14]. Though various solutions have been proposed for the intelligent identification of Benggang, it remains a challenge to construct the optimal data representation and feature-fusion method based on DOMs and DSMs to enhance the algorithm’s identification accuracy.

These studies indicate that, in the field of gully erosion identification research, traditional methods often fail to effectively exploit complementary information from diverse data sources, such as the visual textures of DOMs and the topographic metrics (slope, aspect, curvature, hillshade) derived from DSMs. The resulting weak fusion of heterogeneous features produces data representations that are not robust enough, compromising the model’s ability to distinguish Benggang from non-Benggang regions, especially in areas where subtle surface changes or vegetation obscuration mask critical boundaries. Furthermore, existing models struggle to capture the multiscale spatial and contextual cues essential for identifying complex Benggang landforms, which often exhibit vague edges, varied topographic gradients, and mixed land cover. These limitations lead to poor performance under complex terrain conditions.

To address the aforementioned issues, this paper develops a framework that integrates topographic features derived from DOMs and DSMs with Canny edge detection through a two-stream network with attention mechanisms. This approach resolves the issue of inadequate cross-modal feature integration and enhances the model’s sensitivity to multiscale geometries and visual patterns. The framework enables efficient and automatic gully erosion identification, providing a new technical means for large-scale, low-cost gully surveys and research on gully erosion mechanisms.

## 2. Data Preparation

### 2.1. Study Area

The study area selected for this research is located in Xixi Village, Longmen Town, Anxi County, Fujian Province, at coordinates $118^{\circ }04^{\prime }\;E,24^{\circ }56^{\prime }\;N$. The multi-source image data for this area included DOM and DSM, which were collected in 2023 and 2024 using a DJI Mavic 3E drone (DJI, Shenzhen, China) in mapping aerial photography mode at a flight altitude of 200 m. Using the 3D Maps module of Pix4D software v4.4.9, the collected data were processed through three steps: initial processing, point cloud and texture generation, and DSM orthophoto generation with index calculation, yielding DOMs (resolution: 0.05 m) and DSMs (resolution: 0.05 m) of the study area, as shown in Figure 2. For the purpose of this study, the DOM and DSM images of the entire study area were divided into blocks of approximately 620 × 620 pixels using a grid division method, with the actual area of each block about 31 m × 31 m. During the registration process, the two types of images were cropped into tiles of identical size to ensure spatial correspondence between the corresponding regions.

The number of blocks in each area is summarized in Table 1, and 70% of the data were allocated for the training set and the remaining 30% for the test set to complete the training and testing of the proposed algorithm. In Table 1, it was found that the number of Benggang areas (positive samples) is significantly fewer than that of non-Benggang areas (negative samples). To achieve a balanced representation of classes, we have adopted preprocessing-based data augmentation. Specifically, before training, we applied a series of augmentation operations to the positive samples. We simultaneously performed random horizontal flipping, vertical flipping, rotation, and resizing on both the DOM and DSM images of Benggang regions. The newly generated samples from each transformation were saved to disk, directly increasing the number of positive samples. Random cropping was deliberately excluded from the augmentation pipeline. This exclusion ensured that the critical topographic structures of Benggangs remained entirely intact within the image boundaries, thus preventing the potential loss of their distinctive features. By utilizing the expanded dataset in subsequent training, the model was exposed to a greater variety of positive samples.

### 2.2. Processing of DSM Data

Many scholars have used statistical models, machine learning algorithms, and other data analysis techniques to analyze influencing factors such as terrain, vegetation, landforms, and land use from multiple perspectives [15,16]. Comparative analyses have shown that slope and aspect were significantly positively correlated with Benggangs. Slope primarily determines the steepness of Benggangs and the gradient of the main ditch, while aspect dictates the orientation of Benggangs. Therefore, during the data processing process, the DSM of the area under study was parsed to obtain the features of changes in slope, aspect, hill shade, and curvature. For the convenience of writing, we collectively refer to these four topographic factors as DSM indicators (DSM-I).

Slope: refers to the degree of inclination or steepness of a surface. The slope of any point on the Earth’s ground is the angle between the tangent plane through that point and the horizontal ground. It is calculated as follows: (1)$$\mathrm{Slope}=\arctan\sqrt{f_{x}^{2}+f_{y}^{2}}\times \frac{180}{\pi }$$
where $f_{x}$ and $f_{y}$ are the slope gradient values in the *X* and *Y* directions, respectively.

Aspect: refers to the orientation of the slope and indicates the direction of the steepest slope at a given point on the surface. Aspect can be considered as the direction of the slope that the mountain faces or the direction of the compass. It is calculated as follows: (2)$$\mathrm{Aspect}=\arctan\left(\frac{f_{y}}{f_{x}}\right)$$

Curvature: is a quantitative measure of the degree of distortion of the terrain surface change and refers to the aspect change rate at any point on the surface. It is an indicator that reflects the degree of curvature of contour lines and is calculated as follows: (3)$$\mathrm{Curvature}=Z-\nabla^{2}Z$$
where $\nabla^{2}Z$ is the Laplace operator of the elevation *Z*.

Hill shade: Simulates the shading effect of terrain relief under sunlight, reflecting the micro-topographic changes of the surface. It is calculated using the following formula: (4)$$\mathrm{Hill}\;\mathrm{shade}=255\times \left(\cos\theta_{s}\cos S+\sin\theta_{s}\sin S\cos\left(A-\phi_{s}\right)\right)$$
where $\theta_{s}$ is the solar zenith angle (the angle between the sun’s incident direction and the vertical line), $\phi_{s}$ is the solar azimuth angle (the horizontal angle of the sun’s direction from the north), *S* represents the slope of the terrain (calculated as in Equation (Equation 1)), and *A* represents the aspect of the terrain (calculated as in Equation (Equation 2)). This formula integrates the geometric relationship between the sun’s position and terrain features to generate grayscale values that visually enhance terrain undulations.

Based on the above indicators, this study draws on the work of Shen et al. [17], constructing a deep learning-based feature fusion network to integrate different types of data for intelligent Benggang identification. This approach addresses the limitations of relying solely on DOM visual features in identification. Simultaneously, we construct an end-to-end Benggang identification algorithm by directly performing feature extraction and fusion on DOMs and DSMs, which simplifies data processing workflows and enhances the efficiency of the identification algorithm.

### 2.3. Processing of Edge Data

The formation of Benggangs is typically accompanied by surface destruction and changes, often characterized by distinct boundary lines at the interfaces between different land features. For instance, when Benggangs occur, boundaries that differ from the surrounding environment can be identified through the analysis of edge features on images, thus enabling the initial separation of the target area from the background. To extract edge features from DOM data, we used the Canny operator, a classical edge detection algorithm designed to identify significant edges in images [18]. The processing flow included the following steps: applying Gaussian filters to remove noise, calculating gradients to determine edge intensity and direction, conducting non-maximum suppression to precisely locate edges, applying double-threshold processing to distinguish strong and weak edges, and finally forming a complete boundary through edge connection. The edge visualization of the actual Benggang area is shown in Figure 3.

For the above feature indicators, this study used DOM data as the first branch input of the two-stream fusion network to extract image-based features. The edge detection indicators, DSM, and related terrain feature indicators processed from the data were stacked into five-dimensional indicator data as the second branch input of the two-stream fusion network. Finally, we normalized the data before feeding them into the model. For RGB-type DOM data, we applied the same normalization method as used in the ImageNet dataset. For indicator data, the ranges vary: the slope range is [0,π/2], the aspect range is [−π,π], the curvature value range is [−200,200], the hill shade value is adjusted to [0,255], and the Canny edges’ range is [0,255]. All these values were uniformly normalized to the range [−1,1].

## 3. Methods

### 3.1. Overall Network Architecture

The network architecture of this study, illustrated in Figure 4, takes DOM data and indicator data as inputs to different branches of the two-stream fusion network. Following data augmentation and regularization, the encoder of the model performs deep feature extraction on data from different branches by building multiple layers of Encoder Blocks (EBs), designed to learn complex features. After processing four EBs, the model generated four different-scale features.

At the same time, for different-scale DOM and DSM features extracted by EBs, a layer of feature fusion blocks (FBs) is constructed after each layer of EBs to fuse the multiscale features of the two branches. FBs receive the abstract features extracted by the EBs of the two branches and adopt and fuse two different attention mechanism blocks. Additionally, FBs also receivethe fused features of the previous layer to combine the information from different feature extraction levels.

To further enhance feature representation, FBs employ convolutional kernels with different receptive field sizes and attention mechanisms to capture multiscale semantic information, enabling the model to obtain a more comprehensive feature representation. After feature extraction, abstract features of different branches are passed through the global average pooling (GAP) layer and the linear layer to output the classification results, and each branch is individually predicted with a classification tensor. Finally, in the decision fusion block, the final Benggang classification is obtained after the prediction results are spliced and then passed through the linear layer.

### 3.2. Feature Extraction

ResNet [19] is a landmark achievement in deep learning, which solves the problems of gradient vanishing and gradient explosion in deep network training by introducing residual connections. This breakthrough inspired numerous enhancements based on the ResNet architecture. SE-Net [20] introduces a channel attention mechanism that compresses channel statistical information through global pooling. This mechanism simulates the inter-dependencies between feature map channels and uses global context information to selectively emphasize or suppress features. ResNeXt [21] uses grouped convolutions in ResNet’s bottleneck block to convert multi-path structures into unified operations. SK-Net [22] introduces feature map attention between the two network branches. RegNet [23] is an architectural approach to network design through the search space and outperforms EfficientNet [24]. ResNeSt [25], through an SE-like block, introduces a spatial attention mechanism that can adaptively recalibrate channel feature responses, thus enabling the network to focus more on important feature channels while suppressing less important ones. Furthermore, ResNeSt implements multi-path representations through the Split-Attention block. This structure allows the network to learn independent features from different branches and establishes connections between branches with special attention mechanisms, capturing richer and more diverse feature representations.

Building on these advancements, we adapt ResNeSt as the backbone network for feature extraction in both branches. The design of the Encoder Block (EB) is shown in Figure 5. In this structure, the input feature map is divided into multiple cardinal groups, and each cardinal group is further subdivided into multiple splits. Each split undergoes feature transformation through 1 × 1 and 3 × 3 convolutional layers. The Split-Attention block then integrates these features using global average pooling and attention weight calculation to emphasize significant features while suppressing less important ones. Finally, the outputs of all splits are spliced and passed through a 1 × 1 convolutional layer for residual connections with the original inputs, forming the output features for each layer of the Encoder Block.

The features of different input data types are extracted by stacking multiple EB layers (four layers are used in this study) as follows:(5)$$F_{i}^{1}=EB_{i}\left(F_{i-1}^{1}\right),F_{0}^{1}=DOM$$(6)$$F_{i}^{2}=EB_{i}\left(F_{i-1}^{2}\right),F_{0}^{2}=Indicator\;Data$$
where i∈[1,4] denotes the current number of EB layers, and $F_{i}^{1}$ and $F_{i}^{2}$ denote the feature output of the first branch and the second branch, respectively. Using smaller convolutional kernels (e.g., 1 × 1) in the EB layer helps retain localized information, while stacking additional EB layers increases the network’s receptive field, enabling it to better capture high-level features from images.

### 3.3. Fusion Block

In recent years, many studies have adopted attention mechanisms and feature fusion techniques to enhance model performance in image identification tasks. Woo et al. [26] proposed the Convolutional Block Attention Module (CBAM), a simple and effective attention module that sequentially infers channel attention (CA) and spatial attention (SA) maps, multiplying them with the input feature maps to adaptively refine the features. Subsequently, Zhang et al. [27] grouped the channel dimensions and used the Shuffle Unit to construct both channel and spatial attention for each sub-feature, achieving information interaction via the “channel shuffle” operator. Liu et al. [28] proposed Polarized Self-Attention (PSA) for pixel-level regression, using polarization filters to maintain high resolution during the calculation of channel and spatial attention. Tang et al. [29] fused complementary information across modalities by calculating channel attention weights, weighting complementary features, and adding them to the original features during infrared and visible image fusion. Huo et al. [30] significantly improved the classification accuracy in medical image fusion by splitting the channels and spatial attention mechanism of CBAM for fusion.

Building on these approaches, we proposed a feature fusion block that combines CA and SA mechanisms to integrate and refine DOM and indicator data features across different scales, as shown in Figure 6. (C, H, W) represents the number of feature channels, C, and the height and width are H and W, respectively. The feature fusion layer processes feature inputs from three sources and fuses multiscale information by stacking multiple FB layers.

The first input consists of image branch features $F_{i}^{1}$ extracted from Benggang images, where *i* denotes the encoder’s layer count. These image features contain rich visual information on the surface features of Benggangs (e.g., collapse walls, erosion gullies). Using the spatial attention mechanism, these features are processed to generate $SA(F_{i}^{1})$, enabling the network to adaptively focus on key feature areas on images, such as areas with significant features of Benggangs, thereby enhancing feature representation.

The second input includes indicator data branch features $F_{i}^{2}$, which are initially processed to obtain $CA(F_{i}^{2})$ with the channel attention mechanism. After global max pooling and average pooling operations, the global maximum and average features are extracted. These two types of features are then fed into a shared MLP (multilayer perceptron), which generates channel-wise weight vectors by performing dimensionality reduction and expansion to capture inter-channel dependencies. Finally, the weights are normalized to the range [0, 1] via Sigmoid activation and applied to the original feature maps through element-wise multiplication, enabling adaptive weighting and recalibration of channel-wise features to emphasize informative channels and suppress irrelevant ones.

The third input consists of features from the previous FB layer, $FB_{i-1}$, which are processed through convolution and average pooling to generate $FB_{i-1}^{\prime }$ (for the first layer, $FB_{i}$ only receives the first two inputs). These features are then concatenated with $F_{i}^{1},F_{i}^{2}$ to produce $FB_{i-1}^{\prime \prime }$ by layer normalization and convolution operation. Layer normalization stabilizes network training, while convolution operations extract and integrate features. The calculations are as follows:(7)$$\left\{\begin{array}{cc} SA(F_{i}^{1}) & =\sigma (Conv_{7\times 7}\left(Concat\left[AvgPooling\left(F_{i}^{1}\right),MaxPooling\left(F_{i}^{1}\right)\right]\right))\\ CA(F_{i}^{2}) & =\sigma (MLP\left(AvgPooling\left(F_{i}^{2}\right)+MLP\left(MaxPooling\left(F_{i}^{2}\right)\right)\right))\\ FB_{i-1}^{\prime } & =AvgPool(Conv_{1\times 1}\left(FB_{i-1}\right))\\ FB_{i-1}^{\prime \prime } & =GELU(Conv_{1\times 1}\left(LNorm\left(Concat\left[F_{i}^{1},F_{i}^{2},FB_{i-1}^{\prime }\right]\right)\right)) \end{array}\right.$$
where σ is a Sigmoid function, and Conv(∗) is the convolution operation of various kernel sizes. AvgPool(∗) is the average pooling layer. MaxPool(∗) is the maximum pooling layer. MLP(∗) is a multilayer perceptron with fully connected layers. Concat[∗] is feature concatenation. LNorm(∗) is layer normalization, and GELU is an activation function. LNorm(∗) is batch normalization.

For the features $SA(F_{i}^{1})$, $CA(F_{i}^{2})$, and $FB_{i-1}^{\prime \prime }$ of the three branches mentioned above, $SA(F_{i}^{1})$ and $CA(F_{i}^{2})$ are first combined with the original features $F_{i}^{1},F_{i}^{2}$, respectively, to obtain element-wise multiplication. These results are then concatenated with $FB_{i-1}^{\prime \prime }$ to form the fused feature ${\tilde{F}}_{i}$, which fully integrates multi-source feature information. Finally, the fused feature ${\tilde{F}}_{i}$, after convolution through a residual network and two convolutional layers with a kernel size of 1×1, is then added with $FB_{i-1}^{\prime }$ to obtain $FB_{i}$, the fused feature of the *i*th layer. After passing through four FB layers, the output of the fusion branch $FB_{4}$ is obtained. The calculations are as follows:(8)$$\left\{\begin{array}{cc} {\tilde{F}}_{i} & =Concat[SA\left(F_{i}^{1}\right)⊗F_{i}^{1},CA\left(F_{i}^{2}\right)⊗F_{i}^{2},FB_{i-1}^{\prime \prime }]\\ FB_{i} & =BN\left(Conv_{1\times 1}\left(GELU\left(Conv_{1\times 1}\left(BN\left(GELU\left(Conv_{3\times 3}\left(LN\left({\tilde{F}}_{i}\right)\right)\right)+LN\left({\tilde{F}}_{i}\right)\right)\right)\right)\right)\right)+FB_{i-1}^{\prime } \end{array}\right.$$

### 3.4. Decision Fusion Block

The decision fusion (DF) block integrates the prediction results of different features to produce a final classification label. The model processes input features using three parallel global average pooling (GAP) layers to perform down-sampling. The pooled features are then passed through three fully connected (FC) layers, reducing the feature dimensions sequentially from 2048 to 512, and finally to 1. This process yields three 1-dimensional outputs, o1,o2,o3, calculated as follows:(9)$$\left\{\begin{array}{cc} o1 & =\mathrm{FC}3(\mathrm{FC}2\left(\mathrm{FC}1\left(\mathrm{GAP}\left(F_{4}^{1}\right)\right)\right))\\ o2 & =\mathrm{FC}3(\mathrm{FC}2\left(\mathrm{FC}1\left(\mathrm{GAP}\left(FB_{4}\right)\right)\right))\\ o3 & =\mathrm{FC}3(\mathrm{FC}2\left(\mathrm{FC}1\left(\mathrm{GAP}\left(F_{4}^{2}\right)\right)\right)) \end{array}\right.$$

Subsequently, the three 2048-dimensional outputs from $\mathrm{GAP}\left(F_{4}^{1}\right),\mathrm{GAP}\left(FB_{4}\right),$ and $\mathrm{GAP}(F_{4}^{2})$ are concatenated into a 3×2048 matrix *M*. This matrix is further processed by a 512-dimensional fully connected layer (FC4) to produce o4. The calculations are as follows:(10)$$\left\{\begin{array}{cc} M & =Concat[GAP\left(F_{4}^{1}\right),\mathrm{GAP}\left(FB_{4}\right),\mathrm{GAP}\left(F_{4}^{2}\right)]\\ o4 & =\mathrm{FC}4(M) \end{array}\right.$$

Finally, the four 1-dimensional outputs, o1,o2,o3,o4, are concatenated and passed through a fully connected layer (FC5) to generate the final classification label $\hat{y}$. Here, $FC_{i}$ denotes the *i*th fully connected layer, and [a,b,c] represents the concatenation of vectors *a*, *b*, and *c*. The computation is as follows:(11)$$\hat{y}=\mathrm{FC}5\left(\left[o1,o2,o3,o4\right]\right)$$

### 3.5. Loss Function

To address the imbalance between positive and negative samples in Benggang classification tasks, this study utilized the Binary Cross-Entropy Loss (BCELoss) function with a weighted approach to enhance the model’s sensitivity to the minority class (i.e., Benggang samples). A positive sample weight, $w_{i}$, is defined and set to 5.0, meaning that the loss for each positive sample is magnified by a factor of 5 during the loss function calculation. This approach aims to balance the model’s attention to positive and negative samples during the training process. The BCELoss function is defined as follows:(12)$$\mathrm{BCELoss}=-\frac{1}{N}\sum_{i=1}^{N}\left[w_{i}\cdot y_{i}\cdot \log\left({\hat{y}}_{i}\right)+\left(1-y_{i}\right)\cdot \log\left(1-{\hat{y}}_{i}\right)\right]$$
where *N* is the total number of samples, $y_{i}$ is the true label (0 or 1) for the $i_{th}$ sample, ${\hat{y}}_{i}$ is the predicted probability for the $i_{th}$ sample being positive, and $w_{i}$ is the weight assigned to positive samples.

## 4. Experiment

### 4.1. Evaluation Metrics and Implementation Details

Training schedule. Data augmentation and regularization were performed using the data processing methods described in Section 2. During training, the EB of the dual-stream network utilized ResNeSt with unshared weights, the loss function was BCELoss, and the optimizer was Adam. The number of epochs was set to 100. After completing training, the final performance of different schemes was compared on the test set.

Evaluation Metrics. To evaluate the accuracy of the Benggang identification model, we use accuracy, precision, recall, and F1-score to validate the prediction performance of the model. The difference between the model’s predicted classifications and the actual labels is used to estimate the accuracy of the model. The metrics are defined as follows:(13)$$\mathrm{Accuracy}=\frac{\mathrm{TP}+\mathrm{TN}}{\mathrm{TP}+\mathrm{FP}+\mathrm{FN}+\mathrm{TN}}$$(14)$$\mathrm{Precision}=\frac{\mathrm{TP}}{\mathrm{TP}+\mathrm{FP}}$$(15)$$\mathrm{Recall}=\frac{\mathrm{TP}}{\mathrm{TP}+\mathrm{FN}}$$(16)$$F1=2\times \frac{\mathrm{Precision}\times \mathrm{Recall}}{\mathrm{Precision}+\mathrm{Recall}}$$

Here, TP (true positive) refers to the number of samples correctly predicted as Benggang. TN (true negative) refers to the number of samples correctly predicted as non-Benggang. FP (false positive) refers to the number of samples incorrectly predicted as Benggang. FN (false negative) refers to the number of samples incorrectly predicted as non-Benggang.

### 4.2. Comparison Experiments

To further validate the effectiveness of the proposed algorithm for Benggang identification, we compare the accuracy, recall, and F1-score of MS-TSFN with several state-of-the-art deep learning models. In the comparison experiments, all models used a combination of DOM and indicator data as input samples. As shown in Table 2, the experiment includes the evaluation metrics accuracy (Acc), precision (Prec), recall, and F1-score.

As shown in Table 2, MS-TSFN demonstrated significant advantages in all evaluation metrics related to Benggang classification. Specifically, the F1-score of MS-TSFN improves by 0.1058 (0.8095 vs. 0.7037) and 0.1291 (0.8095 vs. 0.6804) compared to ResNeSt and Deit-B, respectively. Similarly, the accuracy of MS-TSFN increased by 5.88% (92.76% vs. 86.88%) and 5.43% (92.76% vs. 87.33%) compared to MambaOut and Eva02, respectively. These results indicate that the integration of a two-stream convolutional neural network and feature fusion techniques significantly enhances Benggang classification performance.

### 4.3. Analysis of Different Data Types

We utilize the well-performing ResNeSt model to compare and analyze classification performance using single data types. Table 3 shows the performance when each type of data is input individually, where DSM-I includes four additional feature indicators: slope, aspect, hill shade, and curvature. DOM data produces the best results across all metrics, achieving an accuracy (Acc) of 87.78%, recall of 84.09%, and F1-score of 0.7327. DOM images provide rich information such as surface texture and color, which facilitates the identification of Benggang surface features and significantly enhances the model’s classification performance.

Next, the Canny edge features show a substantial decline compared to DOM, with accuracy, precision, recall, and F1-score decreasing by 23.53%, 33.34%, 15.91%, and 0.3010, respectively. This indicates that the Canny edge model struggles to classify Benggang areas when relying solely on edge detection results. Finally, DSM-I, even without the rich image features of DOMs, achieve respectable performance, with accuracy, precision, recall, and F1-score of 85.07%, 60.81%, 70.45%, and 0.6526, respectively. These results suggest that DSM-I captures unique Benggang characteristics, which the model can effectively extract.

We also analyze the impact of different data combinations on Benggang identification. The results, shown in Table 4, indicate that the combination of DOM, DSM-I, and Canny (Row 3) outperformed DOM and DSM-I alone (Row 2), improving accuracy, precision, recall, and F1-score by 3.17%, 10.60%, 4.54%, and 0.0739, respectively. These findings suggest that Canny, when combined with DOM and DSM-I, overcomes its inherent limitations by enhancing edge information. Benggangs, as a complex erosional landform with unclear boundaries, often have edges around their walls and channels that are not clearly visible in DOM images. Canny edge detection enhances edge features, enabling the model to more accurately delineate the extent of Benggang areas.

Furthermore, the combination of DOM and Canny (Row 1) is inferior to DOM and DSM-I (Row 2) across all metrics. DSM-I provides critical information about elevation changes, depth, slope, and other topographic features of Benggangs. When used alone, DSM-I lacks surface detail information, which can impact identification accuracy. However, combining DSM-I with DOM yields more accurate results. The combination of all three data types—DOM’s visual information, DSM-I’s topographic data, and Canny’s edge features—complements each other, capturing Benggang characteristics from multiple dimensions. This synergy improves the model’s ability to identify and analyze Benggang areas, enhancing all evaluation metrics.

### 4.4. Analysis of Network Blocks

Table 5 summarizes the performance of models with different backbone network layer configurations. Each element in the Num_layer list corresponds to a residual block group, and the value of the element represents the number of residual blocks in that residual block group. For example, the list [3, 4, 6, 3] contains four elements, which indicates that the entire ResNeSt network has four main residual block groups. It means that the first residual block group contains three residual blocks. The second residual block group contains four residual blocks. The third residual block group contains six residual blocks. The fourth residual block group contains three residual blocks. Compared to the layer configuration of [1, 1, 1, 1], the model with a [2, 2, 2, 2] configuration shows slight decreases in accuracy and F1-score, but its recall increased significantly by 13.64%. This result suggests that deeper network structures improve the model’s ability to recognize positive samples. When the layer configuration is set to [3, 4, 6, 3], the accuracy for Benggang identification reaches 92.76%, and the F1-score reaches 0.8095, showing the best classification performance in the context of class imbalance between Benggang and non-Benggang samples.

This section also compares the classification performance of Benggang and non-Benggang areas using different feature fusion modes. Table 6 shows the model performance with various attention mechanisms applied to the fusion block. AM1 (Attention Mechanism 1) refers to the attention mechanism applied to the first input (RGB data) in the fusion block (FB), while AM2 (Attention Mechanism 2) is applied to the third input (indicator data). The SAM-CAM configuration achieves the highest accuracy (92.76%), improving by 3.17% over CAM-CAM (the lowest accuracy). SAM-CAM also outperforms other configurations in F1-score (0.8095), with an improvement of 0.0679 over CAM-CAM (0.7416, the lowest). The recall of PSA_C-PSA_S is the highest (81.82%), 6.82% higher than CAM-CAM (75.00%, the lowest).

Overall, SAM-CAM delivers the best performance, highlighting the complementary roles of spatial features from RGB data and channel features from indicator data for Benggang classification. The simpler spatial attention mechanism in SAM-CAM yields excellent results, suggesting that the model emphasized spatial features in the image data while relying on channel-based adjustments for indicators. By fine-tuning the weights of less important factors, CAM improves the influence of indicators, resulting in better overall performance.

### 4.5. Decision Fusion (DF) Ablation Study

As shown in Table 7, the decision fusion (DF) block significantly impacts model performance. The accuracy increases by 3.17%, and the F1-score improves by 0.0568, demonstrating the positive optimization effect of DF on the model. The morphological features of Benggangs vary greatly at different developmental stages. For instance, during the early development stage, Benggang boundaries are often vague, and vegetation coverage is high, making it challenging to distinguish Benggangs from the surrounding environment. By leveraging fused features and multi-stream data results, DF enhances the model’s ability to identify key features, improving its overall classification performance.

### 4.6. Analysis of Different Map Tile Sizes

Table 8 compares model performance across different tile sizes. The first row shows the results of Benggang classification with 7 m × 7 m tile sizes, which involves 17,756 images, including 1965 images of Benggang areas and 15,791 images of non-Benggang areas. The second row displays the results for 31 m × 31 m tile sizes. Compared to the 31 m × 31 m tiles, the 7 m × 7 m tiles resulted in a 6.39% decrease in accuracy, a 19.11% decrease in recall, and a 0.3244 drop in F1-score.

This significant decline suggests that the smaller 7 m × 7 m tiles provide insufficient geographical information. With smaller tiles, DOM data becomes more challenging to differentiate because vegetation often obscures Benggang areas, leading to their misclassification as non-Benggang areas. Additionally, the height variations captured by DSM within a smaller range are insufficiently distinct to assist the DOM data in accurate Benggang classification. In contrast, the 31 m × 31 m tile size performed better because larger tiles include more comprehensive geographical information, which is critical to the model learning process.

Although larger tiles may result in less precise classification, they provide more geographical information, which is crucial for the model’s learning process.

### 4.7. Heatmap Visualization Analysis

Figure 7 shows heatmaps visualized during the Benggang identification process, illustrating the contribution distribution to the prediction output. Darker red areas indicate higher scores, signifying regions in the original image that elicit stronger network responses and contribute more significantly to the prediction.

From the comparative analysis of the images, the DOM branch clearly separates vegetated areas from exposed regions. As shown in the first and second rows of Figure 7, the model focuses more on non-vegetated areas in the DOM branch, particularly on white exposed rocky regions where more detailed information was extracted. In contrast, the indicator branch concentrates on areas with significant topographic variations, such as higher altitudes and steeper slopes. Even in vegetated areas, the network successfully captures these variations.

In the fusion branch of MS-TSFN, the areas of interest from both branches are integrated, and their weights are reallocated. In the fourth column, the fusion results effectively highlight the Benggang areas. In the third row, the DOM branch struggles to focus on large exposed rocky areas within the Benggang, while the indicator branch captures these steep variations. After fusion, the model successfully identified the Benggang areas.

Overall, the areas of interest between the DOM and indicator branches partially overlap but are not entirely identical. While the DOM branch excels in distinguishing vegetated and exposed areas, Benggangs can also occur in vegetated regions. The indicator branch, using DSM data, effectively complements the DOM branch by identifying topographic variations, enhancing the model’s overall identification performance.

### 4.8. Result Visualization Analysis

In this study, visualization is performed on Areas 2, 3, and 4 using the optimal identification settings, and the model’s identification results are shown in Figure 8. The main Benggang areas have been successfully identified in all three regions. Most misidentified areas occur at the edges of Benggang regions, where tiles are misclassified as non-Benggang features. However, these misclassifications have minimal impact on accurately locating the actual Benggang areas. Instances of non-Benggang areas being misclassified as Benggangs are relatively rare.

Overall, the results demonstrate the model’s exceptional ability to process geographic information and accurately identify Benggang areas in complex environments.

## 5. Discussion

The proposed MS-TSFN model addresses the limitations of single-data Benggang classification by leveraging multi-source fusion of DOM- and DSM-derived topographic features and and Canny edge detection, significantly improving identification accuracy compared to methods relying on isolated data types.

In the field of Benggang erosion analysis, numerous studies have conducted detailed investigations into influencing factors using statistical analysis methods, information quantity models, machine learning algorithms, and other approaches. For example, slope and aspect, as primary topographic indicators, significantly impact the model’s ability to identify Benggang-prone areas [9,37]. Steeper slopes enhance gravitational and hydrodynamic forces, accelerating soil erosion and collapse—core mechanisms of Benggang formation. The model utilizes slope (calculated via Equation (Equation 1)) to distinguish unstable terrain, while aspect (Equation (Equation 2)) captures directional erosion patterns, such as south-facing slopes in subtropical regions experiencing more intense rainfall erosion.

Curvature and hillshade further refine micro-topographic details. Curvature influences flow convergence and divergence, affecting soil erosion processes, while hillshade emphasizes concave/convex surface variations and light/shadow contrasts critical for delineating erosion boundaries (e.g., collapse slopes and debris accumulation zones) [38]. Additionally, Wei et al. [1] defined a Benggang as a large-scale gully. Studies [1,15,39] have explored the impacts of other factors on gully erosion susceptibility, including rainfall erosivity, soil moisture, and precipitation, among others. This represents a future direction for integrating natural factors with multi-source data.

Based on the principles of data availability and operability of image factor extraction, slope, aspect, curvature, and other factor data were ultimately selected as input for feature extraction. Through deep learning, the model achieves synergistic integration of DOM’s visual textures (e.g., exposed soil and gully patterns) and DSM’s topographic gradients (slope, aspect), which together highlight erosion-prone habitats. Therefore, the proposed method effectively discriminates between Benggang and non-Benggang regions across diverse scenarios, correctly identifying typical Benggang morphologies and distinguishing non-erosion areas with complex backgrounds.

However, observations from the study results reveal persistent misclassifications in challenging environments, with potential influencing factors as follows: (1) The model remains susceptible to identification biases caused by terrain, shooting angle, lighting, and vegetation—for example, hill shade or dense vegetation masking erosion boundaries, which disrupts edge detection and topographic calculations. (2) Scale-dependent ambiguities, where small Benggang patches within large non-Benggang tiles are misidentified due to insufficient contextual information in smaller image blocks. These issues highlight the trade-off between spatial resolution and contextual awareness.

To address these limitations, future studies could follow three key directions. First, expanding the dataset to include multiregional and multiseasonal data, while incorporating additional sources such as LiDAR, multispectral imagery, or soil moisture data, would enhance the model’s adaptability to heterogeneous environments. Second, exploring self-supervised or weakly supervised learning strategies could reduce reliance on manual feature engineering and labeled data, which is particularly valuable for scenarios with scarce Benggang samples. Third, developing adaptive fusion strategies will address residual errors in complex terrain.

## 6. Conclusions

This study tackles the challenge of accurately classifying Benggangs using drone-derived DOM and DSM data, where traditional methods struggle to fuse visual and topographic features. We propose the MS-TSFN, which uses a ResNeSt backbone to extract features from DOM- and DSM-derived indicators and Canny edges. An attention-based fusion module emphasizes spatial details in DOMs and topographic correlations in DSMs, while a decision fusion block enhances prediction robustness. By leveraging cross-modal associations and multiscale learning, MS-TSFN improves identification accuracy in complex terrains. Experiments in Fujian Province show it achieves 92.76% accuracy and a 0.8095 F1-score, surpassing state-of-the-art methods by combining visual, topographic, and edge features. Future work will focus on expanding dataset diversity, improving boundary segmentation, and exploring advanced learning strategies for better generalizability and ecological applications.

## Figures and Tables

**Figure 1 sensors-25-02924-f001:**
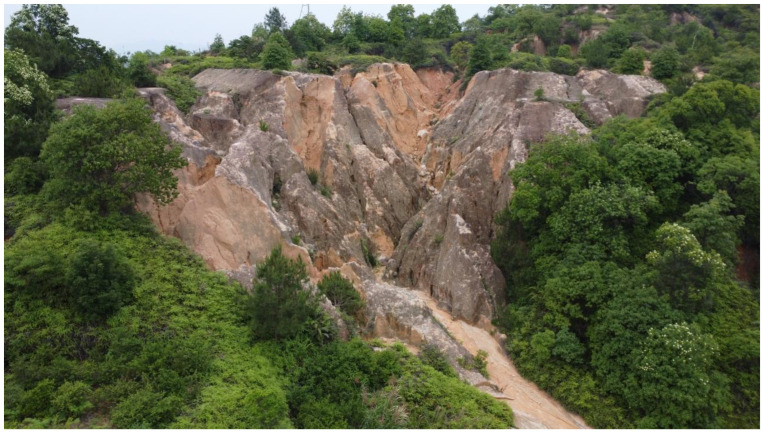
A typical Benggang in the study area.

**Figure 2 sensors-25-02924-f002:**
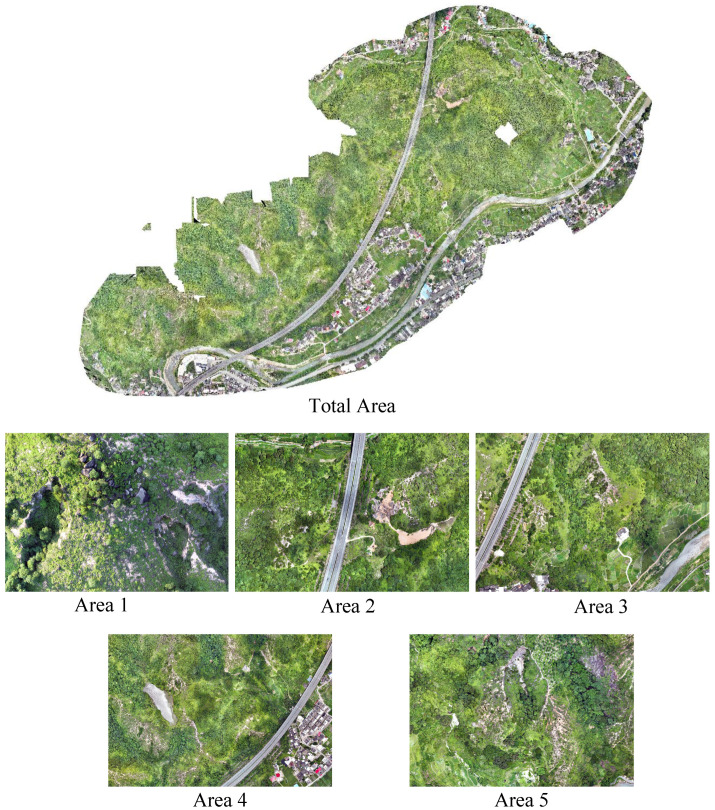
Benggang images collected.

**Figure 3 sensors-25-02924-f003:**
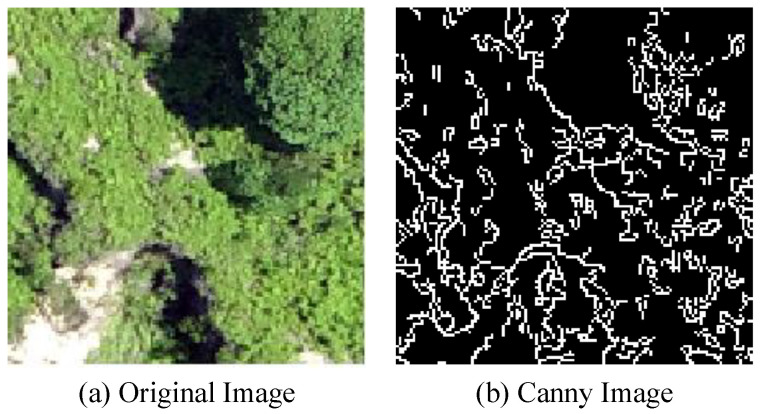
Boundary visualization of Canny operator.

**Figure 4 sensors-25-02924-f004:**
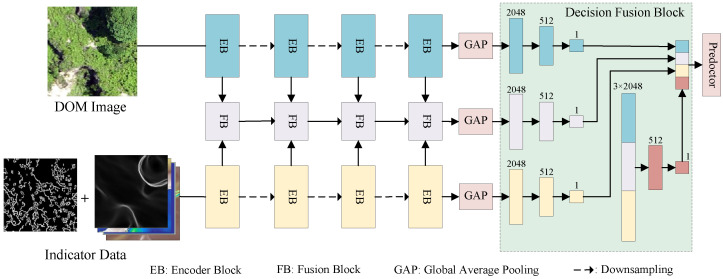
Benggang identification framework of MS-TSFN.

**Figure 5 sensors-25-02924-f005:**
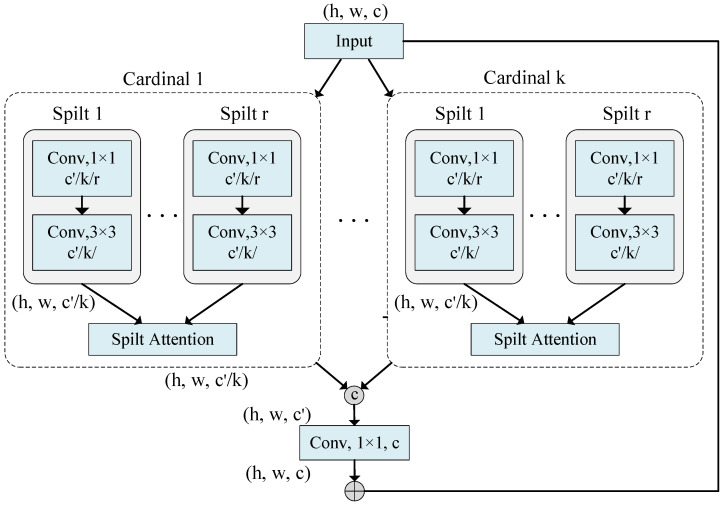
Structure of Encoder Block.

**Figure 6 sensors-25-02924-f006:**
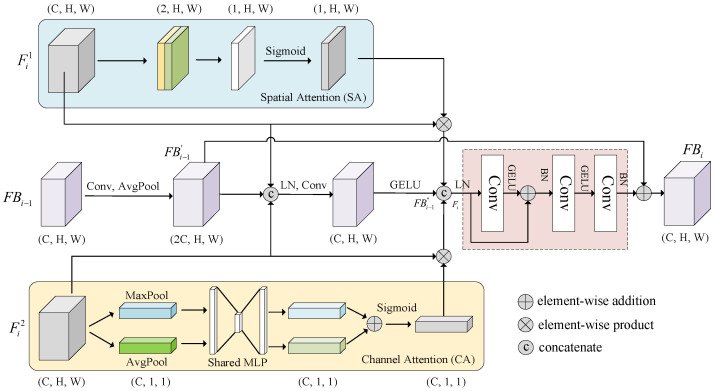
Fusion block.

**Figure 7 sensors-25-02924-f007:**
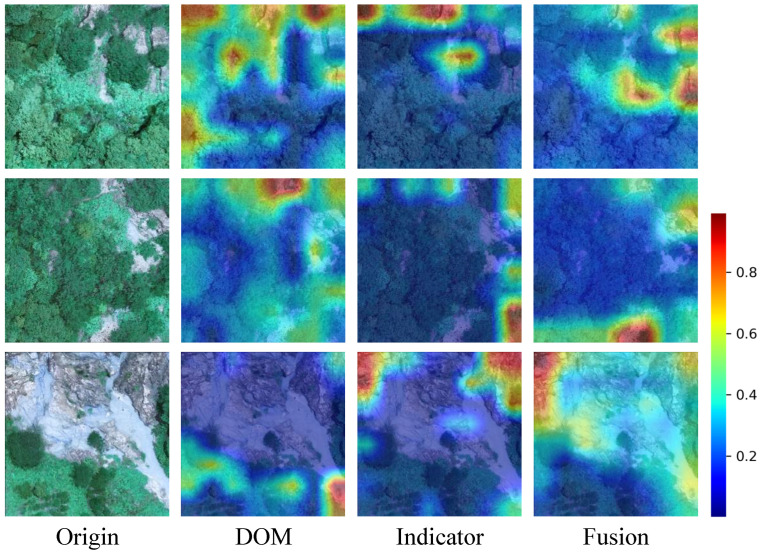
Heatmaps visualization for Benggang identification. The first column represents the original images, the second column shows the heatmaps from the last layer of the DOM branch, the third column displays the heatmaps from the last layer of the indicator branch, and the fourth column represents the heatmaps from the last layer of the two-stream FB branch.

**Figure 8 sensors-25-02924-f008:**
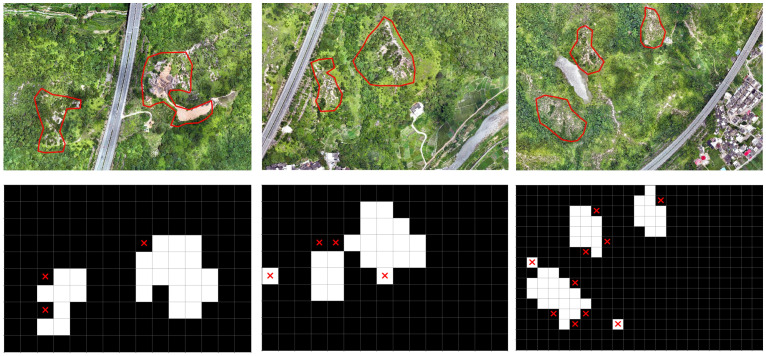
Visualization of the classification of Benggang areas. The original images depict field scenes, with red lines marking the two Benggang areas for comparison with the identification results. Benggang areas are marked in white, non-Benggang areas are marked in black, and misidentified areas are labeled with a red “X”.

**Table 1 sensors-25-02924-t001:** Division of Benggang datasets.

Area	Benggang	Non-Benggang
Area 1	8	4
Area 2	30	120
Area 3	24	126
Area 4	50	318
Area 5	10	50
Total	122	618

**Table 2 sensors-25-02924-t002:** Comparison of popular methods and the proposed model.

Method	Acc (%)	Prec (%)	Recall (%)	F1
ResNet50 [19]	85.97	69.78	52.27	0.5974
Deit-B [31]	85.97	62.26	75.00	0.6804
Swin-B [32]	84.62	58.61	77.27	0.6667
Focalnet [33]	85.52	67.65	52.27	0.5897
Swin_v2 [34]	86.88	68.24	63.64	0.6588
Eva02 [35]	87.33	68.96	65.91	0.6744
MambaOut [36]	86.88	70.27	59.09	0.6420
ResNeSt [25]	92.76	70.37	70.37	0.7037
MS-TSFN	92.76	85.00	77.27	0.8095

**Table 3 sensors-25-02924-t003:** Analysis of Benggang classification with single data types.

Data	Acc (%)	Prec (%)	Recall (%)	F1
DOM	87.78	64.92	84.09	0.7327
Canny	64.25	31.58	68.18	0.4317
DSM-I	85.07	60.81	70.45	0.6526

**Table 4 sensors-25-02924-t004:** Analysis of different data combinations.

Stream1	Stream2	Acc (%)	Prec (%)	Recall (%)	F1
DOM	Canny	84.16	58.51	70.45	0.6392
DOM	DSM-I	89.59	74.40	72.73	0.7356
DOM	DSM-I, Canny	92.76	85.00	77.27	0.8095

**Table 5 sensors-25-02924-t005:** Model performance with different layer configurations.

Num_Layer	Acc (%)	Prec (%)	Recall (%)	F1
[1, 1, 1, 1]	90.50	81.04	68.18	0.7407
[2, 2, 2, 2]	88.24	66.68	81.82	0.7347
[3, 4, 6, 3]	92.76	85.00	77.27	0.8095

**Table 6 sensors-25-02924-t006:** Performance of the model with different fusion blocks.

AM1	AM2	Acc (%)	Prec (%)	Recall (%)	F1
PIA	-	90.50	58.32	77.78	0.6667
CAM	CAM	89.59	73.32	75.00	0.7416
Focal	CBAM	90.05	75.00	75.00	0.7500
Shuffle	CAM	90.95	81.60	70.45	0.7561
PSA-C	PSA-S	90.05	72.01	81.82	0.7660
SAM	CAM	92.76	85.00	77.27	0.8095

**Table 7 sensors-25-02924-t007:** Experimental results of DF.

DF	Acc (%)	Prec (%)	Recall (%)	F1
×	89.59	71.43	79.55	0.7527
✔	92.76	85.00	77.27	0.8095

**Table 8 sensors-25-02924-t008:** Performance of the model with different tile sizes.

Size	Acc (%)	Prec (%)	Recall (%)	F1
7 m × 7 m	86.37	41.62	58.16	0.4851
31 m × 31 m	92.76	85.00	77.27	0.8095

## Data Availability

The data presented in this study are available on request from the corresponding author due to privacy concerns and legal restrictions associated with geographic data.

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
