# Peer review of "Multiscale Two-Stream Fusion Network for Benggang Classification in Multi-Source Images"

_sensors, 2025, doi:10.3390/s25092924_

Round 1
Reviewer 1 Report
Comments and Suggestions for Authors
The authors propose in the paper a method for the classification of zones with the presence of Benggang. The method processes data in two branches, using as input for the first branch data and for the second branch a combination of edges, DSM and terrain feature indicators; at the same time, the partial outputs of these two branches are merged. According to the authors, the classification model is based on ResNeSt. The three outputs are processed through dense layers to finally go to a two-class classification layer.
Several aspects of the paper need to be revised and corrected:
Major issues:
1. The main issue has to do with the organization and scope of what is proposed in the paper. The first part implies that the proposal is about the classification model, but then the results show a good number of comparisons that generate confusion with respect to what was intended to be achieved. In fact the conclusions start by saying that the study investigated the relationship between natural factors and Benggang, in particular the influence of slope, aspect, hillshade, curvature and edge indicators on Benggang characteristics.
2. How is the image registration process between DSM and DOM performed?
3. What is the impact of using the method of Shen et al. [16] on the results?
4. Line 162: Do you apply data augmentation and regularization? How did you apply it?
5. It is necessary to provide details about the training of the complete model shown in Fig. 3. Does it correspond to a single model, or are there several, for how many epochs it was trained, etc.
6. Line 244: “and then processed through the channel attention mechanism”. There is no information about the processing applied here.
7. It is not clear where the MLP referred to in line 255 is located in Fig. 5.
8. I think it is necessary to add more details to Figure 5, such as the names or functions of some blocks, or the dimensions.
9. Specify in Fig. 3 which is the Decision Fusion Block.
10. Add the Precision Equation
11. Lines 296-299: specify which class was taken as positive and which class as negative.
12. Add the precision results in the tables
13. It is necessary to specify what the layer configuration of [x,x,x,x,x] refers to.
14. Update the conclusions according to the scope of the article.
Comments on the Quality of English Language
Minor issues:
1. Line 133: to reduces the data -> to reduce the data
2. Line 134: and improves the computation efficiency. This study uses -> and improve the computation efficiency, this study uses
3. Line 212: "Using smaller convolutional": smaller regarding what?
Reviewer 2 Report
Comments and Suggestions for Authors
The research method in this paper is novel, which has great application value for the extraction and treatment of Benggang in southern China. After comprehensive consideration, the opinions of major review are given, mainly in the part of quotation and discussion. As follows:
1.Introduction
The last part should mention what problems this study has solved, not what contributions it has made.At the same time, it is suggested to add typical pictures of Benggang to illustrate several components of collapse. Because sensors is an international journal, readers need to know and be clear about the research object of this paper.
2.Discussion
There is no discussion part in this article, so it is suggested to add it. For example, the influence of slope, shape and vegetation on the research results of this paper. Moreover, in the discussion part, the shortcomings of this paper and the direction of further research should be added.
Round 2
Reviewer 1 Report
Comments and Suggestions for Authors
The authors have improved the document according to the observations presented in the first revision. Certain aspects need to be clarified/improved before possible publication:
- The authors talk about data augmentation to balance class representation. However, in frameworks such as Pytorch or Tensorflow data augmentation techniques do not increase the number of samples of the original dataset, but generate random transformations of the original images. Therefore, how do you use data augmentation to balance the classes?
- The authors state “with the model's performance evaluated on the test set at the end of each epoch”. However, the evaluation of the model during training, i.e. after each epoch, is performed on validation data and not on test data. This aspect should be clarified and consistent throughout the document.
Reviewer 2 Report
Comments and Suggestions for Authors
I think my first question to the author has been solved. So I think it can be accepted in the current state of the paper.
Author Response
Thank you so much for your meticulous review and guidance.